# An Investigation of *Burkholderia pseudomallei* Seroprevalence in Market Pigs Slaughtered at Selected Pig Abattoirs in Uganda

**DOI:** 10.3390/pathogens11111363

**Published:** 2022-11-16

**Authors:** John E. Ekakoro, Arnold Lubega, Edrine B. Kayaga, Dickson Ndoboli, Andrew P. Bluhm, Eddie M. Wampande, Jason K. Blackburn, Karyn A. Havas, Michael H. Norris

**Affiliations:** 1Department of Public and Ecosystem Health, College of Veterinary Medicine, Cornell University, Ithaca, NY 14853, USA; 2School of Veterinary Medicine and Animal Resources, College of Veterinary Medicine, Animal Resources and Biosecurity, Makerere University, Kampala P.O. Box 7062, Uganda; 3Spatial Epidemiology and Ecology Research Laboratory, Department of Geography, College of Liberal Arts and Sciences, University of Florida, Gainesville, FL 32611, USA; 4Emerging Pathogens Institute, University of Florida, Gainesville, FL 32601, USA

**Keywords:** pigs, swine, melioidosis, ELISA, neglected tropical disease, *Burkholderia pseudomallei*

## Abstract

*Burkholderia pseudomallei* is a Gram-negative bacterium that causes melioidosis, a disease of humans and animals. It is primarily transmitted through direct contact with contaminated soil and surface water. The epidemiology of this pathogen in Africa, including Uganda, is largely unknown. The objectives of this study were to estimate the seroprevalence of *B. pseudomallei* in pigs slaughtered in central Uganda and to identify potential hotspots for this pathogen in the country. A total of 1035 pig sera were analyzed for serological responses to *B. pseudomallei* with type A and type B LPS using OPS type A and OPS type B ELISAs. Of the 1035 samples, 75 (7.25%, 95% CI: 5.8–9%) were seropositive to the OPS-A ELISA using a two standard deviations (SD) cutoff and 19 (1.84%, 95% CI: 1.2–2.9%) at 3 SD. For the OPS-B ELISA, 93/1035 (8.99%, 95% CI: 7.4–10.9%) were seropositive at the 2 SD cutoff, and 28/1035 (2.71%, 95% CI: 1.9–3.9%) at the 3 SD cutoff. Pigs slaughtered in central Uganda were exposed to *B. pseudomallei,* and there is a higher seroprevalence in the rainy months. Public health awareness campaigns about melioidosis may be needed.

## 1. Introduction

*Burkholderia pseudomallei* is a Gram-negative bacterium that causes melioidosis in humans and animals and has a wide host range, including birds, tropical fish, and reptiles [1]. It is an environmental and saprophytic bacterium that is found in wet soil, vegetation, and surface water in endemic areas [2,3,4]. Transmission of this pathogen to animals and humans occurs primarily via environmental exposure through direct contact with contaminated soils and surface water [5,6,7]. *B. pseudomallei* is highly pathogenic [8] and causes a wide range of clinical presentations across species [9,10,11,12], often showing clinical signs similar to those seen in other diseases like tuberculosis [13]. 

In humans, melioidosis may present as an acute pulmonary infection, acute septicemia, skin lesions, and abscessation in different organs [14]. Pigs, sheep, and goats are commonly affected livestock [15]. In pigs, the disease is usually chronic, manifesting with swollen lymph nodes and abscesses in multiple organs, and such pathology leads to economic losses from the condemnation of the carcasses [15,16]. Often, the disease goes unnoticed until an animal is sent to market. A random tracheal swabbing of pigs in Vietnam revealed asymptomatic carriage of *B. pseudomallei* in approximately 0.88% of pigs tested, an isolation percentage that would equate to approximately 194,000 pigs with active *B. pseudomallei* carriage at the national level [17]. 

Melioidosis is a serious threat to public health that has significant impacts on poor and rural communities [18]. A study from 2015 estimated that the global burden of melioidosis resulted in 4.6 million disability-adjusted life years [19]. Treatment of human infections consists of an intensive therapy phase to control acute infection and a 12-week multi-drug eradication phase to prevent reinfection or chronic infections [20]. This is exacerbated by antimicrobial resistance in most clinical strains of *B. pseudomallei* [1]. Those involved in agriculture face additional risks as infected livestock are a threat to human health through zoonotic and environmental transmission [21]. Farmers are also affected by economic losses from livestock mortality associated with melioidosis. It is evident why *B. pseudomallei* is thought to have potential as a bioweapon given its highly pathogenic nature to humans and animals, low infectious doses, resistance to many drugs, ease of aerosolization, and historical use in warfare of its close genetic relative *Burkholderia mallei* [4,11,22,23]. 

The epidemiology of melioidosis in Africa is almost entirely unknown, creating the impression that the disease has a rare occurrence in this region [22,24]. This lack of evidence regarding disease occurrence in African countries makes its diagnosis, prevention, and control in both humans and animals neglected. The study of *B. pseudomallei* in Africa is important as it is a neglected tropical pathogen that is often misdiagnosed due to a lack of awareness and the limited availability of diagnostic tools [25]. The scientific community has called for studies to investigate the burden of this disease in both animals and humans in Africa [24,26]. 

One study identified Uganda as a priority country where melioidosis was probably endemic [21], and a preliminary study conducted in Uganda in the early 1970s suggested that *B. pseudomallei* was present [27]. Sporadic human melioidosis cases have been reported in Uganda [28], proving its existence, while the extent of the disease in animals and humans remains unknown [22]. Therefore, the specific objectives of this study were to (i) estimate the seroprevalence of *B. pseudomallei* in pigs slaughtered at selected pig abattoirs in Uganda to determine the risk in the region and (ii) identify the potential *B*. *pseudomallei* hotspots through animal sentinels.

## 2. Materials and Methods

### 2.1. Sample Acquisition

Pig serum samples were collected from pigs slaughtered at six pig abattoirs located in the Kampala metropolitan area of central Uganda. The slaughterhouses were purposely selected because of their robust and wide catchment area with pigs sourced from different districts and regions of Uganda. The pigs were sampled using a stratified systematic sampling method weighted by the estimated annual slaughter rate per abattoir. Abattoirs with higher numbers of pigs slaughtered per annum had proportionately higher numbers of pigs sampled. Serum was collected as part of sample collection in support of a separate project, and the remaining serum was analyzed for this study. For serum collection, pigs were given a unique identification tag, and serum was collected using jugular or anterior vena cava venipuncture into a 10 mL clotting tube. For each pig, the following data were also analyzed in this study: date, sex of the pig, breed of the pig, and district of origin. The pig breed is described as local indigenous breeds, exotic breeds, which are often pink European pigs, and mixed. The analyzed sera were collected from May 2021 through April 2022. Upon collection, the samples were transported in a cooler with ice packs to the Makerere University Central Diagnostic Laboratory in Kampala, Uganda, at the end of each sampling event and processed immediately.

### 2.2. Serological Testing

The sera were analyzed using an in-house enzyme-linked immunosorbent assay (ELISA) developed at the Emerging Pathogens Institute at the University of Florida and previously described by Norris and others [29]. The type A and type B *B. pseudomallei* OPS antigens were cleaved from purified lipopolysaccharide (LPS) and used in two separate ELISA reactions. For these ELISAs, flat bottom Immulon4 HBX microtiter plates (Fisher Scientific, Waltham, MA, USA) were coated with *B. pseudomallei* antigens OPS-A or OPS-B, sealed using parafilm, frozen at −80 °C and then shipped on dry ice to Makerere University Central Diagnostic Laboratory in Kampala, Uganda where they were stored at 4 °C until used. For each plate, 300 µL blocking solution was added to each well and incubated for 1 h at room temperature. The blocking solution was made of 5% BLOTTO blocking grade non-fat skim milk powder (Santa Cruz Biotechnology, Dallas, TX, USA) and 0.05% Tween 20 in PBS pH 7.4 (PBST; Pierce 20X PBS Tween-20 Buffer; ThermoFisher Scientific, Waltham, MA, USA). After the 1 h incubation, the plates were washed once with 300 µL of blocking solution. Serum samples were diluted 1:1000 in blocking solution, and 200 µL of the 1:1000 serum samples were added to each well. The plates were then incubated at room temperature for 1 h. The plates were inverted to remove the solution and patted dry with paper towels. After that, the plates were washed three times with 300 µL blocking solution, and each time inverting the plates to remove the contents. Afterward, 100 µL of 1:125,000 anti-swine IgG (FC specific)-peroxidase detection antibody (Sigma Aldrich, St. Louis, MO, USA) diluted in blocking solution was added to each well and incubated for 1 h at room temperature. The plates were then washed three times using 300 µL of washing solution (PBST), and 100 µL of freshly prepared substrate (1-step ultra TMB solution; ThermoFisher Scientific, Waltham, MA, USA) was added to each well and incubated for 5–10 min at room temperature. Afterward, 100 µL of 1 M HCL was added to stop the reaction. The absorbance was read at OD_450_ within 30 min of adding the stop solution. Samples were run in duplicate on two separate plates for both type A and B OPS antigen ELISAs.

### 2.3. Serological Data Analysis

Serological data was collated in Microsoft Excel version 16.64 (Microsoft Corporation, Redmond, WA, USA), and data were analyzed and visualized with STATA 17.0 (College Station, TX, USA) and GraphPad Prism 9.4 (San Diego, CA, USA). Frequencies and proportions were used to summarize the data, and confidence intervals were calculated using the Agresti-Coull method. Pearson’s chi-square test was used to test for an association between sex and breed of pig and exposure to *B. pseudomallei,* and the level of significance was set at 0.05. For each antigen tested by ELISA, the results were analyzed separately, and the average of the two absorbance values obtained from the duplicate analyses was calculated. The mean absorbance units were analyzed by population histogram to calculate cutoffs for determining positive populations as previously described [29] and as suggested by the World Organization for Animal Health (WOAH, formerly OIE) [30]. The cutoffs (absorbance thresholds) were first set at a liberal value of two sample standard deviations (SD) and then conservatively at 3 SD from the mean of the suspected seronegative population. Samples with mean absorbance values outside these cutoffs were classified as seropositive. Samples were further divided into two groups based on acquisition during dry and wet seasons. The bimodal seasonality of rainfall in Uganda generally occurs with maxima around March, April, and May, and September, October, and November [31,32]. Samples acquired in these months were considered wet season samples, and all others were from dry seasons. The Shapiro-Wilk test for normality indicated sample absorbance values had a non-normal distribution, so the absorbance values of samples in wet and dry seasons were analyzed by the non-parametric Mann-Whitney U test for ranks.

### 2.4. Mapping Seroprevalence

To map the distribution of cases, we aggregated total positives and the total number of samples provided by the district, which was the highest-resolution spatial unit identified at the time of submission to the abattoirs. Samples reported as from an unknown district were excluded from spatial analysis. For mapping, we considered samples positive for either OPS-A or OPS-B at the 3 SD cutoff. Prevalence was the total positive for either divided by the samples per district. We considered this the most conservative approach to defining a seropositive sample.

Given the range of sample numbers per district, rate smoothing was used to stabilize the variability of crude prevalence related to either low numerators (number of seropositive samples) or low denominators (number of pigs sampled per district) [33]. Smoothing was conducted in GeoDa (version 1.20) using Empirical Bayes Smoothing (EBS), which is described elsewhere [33]. Briefly, EBS aims to reduce rate variability by adjusting the estimates toward the global mean reducing outliers that may overestimate *B. pseudomallei* exposure or appear as false spatial clusters in later analyses; the greatest adjustment is in polygons with lower sample populations. We chose EBS over the spatial Bayes smoothing approach, as several outermost districts were disconnected from most districts sampled, challenging the decision for an appropriate weights matrix to relate to neighboring districts. The raw and smoothed prevalence were box plotted using the *ggplot* package in R version 4.2.1 to illustrate the effect of smoothing [34]. Raw and smoothed prevalence rates were choropleth mapped in Q-GIS version 3.26.3. District-level GIS data were downloaded from the humanitarian data exchange (https://data.humdata.org; accessed on 27 September 2022). A base map of Africa was downloaded from Open.Africa (https://open.africa/dataset/africa-shapefiles; accessed on 5 October 2022). Water body spatial data were downloaded from the World Resources Institute (https://datasets.wri.org/dataset/waterbodies-in-uganda; accessed on 5 October 2022). Smoothed rates were used to describe spatial patterns and for local spatial autocorrelation analysis.

### 2.5. Local Spatial Autocorrelation of Seroprevalence

To test for spatial clustering of *B. pseudomallei* seroprevalence at the district level, we used the Local Moran’s I statistic in GeoDa version 1.2 [33]. The local Moran’s I is written following [35] as:(1)$$I_{i}=Z_{i}\sum W_{ij}Z_{j}$$
where *I_i_* is the statistic for a district *i, Z_i_* is the difference between the EBS seroprevalence in *i* and the mean EBS prevalence rate for all districts, *Z_j_* is the difference between seroprevalence in district *j* and the mean for all districts. *W_ij_* is the weights matrix. In this study, the 1st order queen contiguity was employed. Here, *W_ij_* equals 1/n if a district shares a boundary or vertex and 0 if not. In this study, districts were defined by submissions to abattoirs, and not all districts were represented. Seven districts were not connected to the concentration of districts in south-central Uganda. However, two of those were connected to each other, meeting the requirements of the 1st order queen weights matrix. The remaining five were neighborless and excluded from the analysis by GeoDa. The statistic identifies spatial clusters of like values surrounded by like values, high-high for areas of clustered seroprevalence or low-low where prevalence rates to *B. pseudomallei* were lower than expected. The Local Moran’s I test also identifies spatial outliers, where values in *i* may be high surrounded by low values (high-low outliers), or where low values in *i* are surrounded by high values (low-high outliers). The significance of spatial clusters was determined with a pseudo-*p*-value generated using 999 permutations against a null model of complete spatial randomness [36]. Significant clusters were determined at *p*-value ≤ 0.05, and spatial clusters and spatial outliers were mapped in Q-GIS.

## 3. Results

### 3.1. Pig Sample Characteristics

The 1035 samples were collected from pigs sourced from 43 districts in the four regions of Uganda. Abattoir locations and the number of samples per district are reported in Figure 1, with abattoirs graduated by size to reflect the number of samples submitted per site. Of the 1035 samples tested, 38 did not report district and were excluded from spatial analysis. The remaining 997 cases were assigned to districts. The majority of pigs were sampled at the Lusanja (347; 33.52%) and Wambizi (310; 29.95%) slaughterhouses. Of the 1035 samples analyzed, 566 (54.68%) were from female pigs, 463 (44.73%) from males, and six (0.58%) were from unknown sex; 162 (15.65%) were from local breed pigs, 568 (54.87%) from exotic pigs, 285 (27.53%) from mixed breeds, and 20 (1.93%) were from unknown breeds.

### 3.2. Analysis of Unadjusted Seropositivity

Of the 1035 samples, 75 (7.25%, 95% CI: 5.8–9%) were seropositive to the OPS-A ELISA at the two SD cutoff and 19 (1.84%, 95% CI: 1.2–2.9%) at 3 SD. For the OPS-B ELISA, 93/1035 (8.99%, 95% CI: 7.4–10.9%) were seropositive at the two SD cutoff, and 28/1035 (2.71%, 95% CI: 1.9–3.9%) were seropositive at the 3 SD cutoff. Forty-four of the 1035 (4.25%, 95% CI: 3.2–5.7%) were seropositive for both Type A and B OPS antigen ELISAs at the 2 SD cutoff, and five (0.48%, 95% CI: 0.2–1.2%) at the threw SD cutoff. The absorbance values of all pig samples to the OPS type A and OPS type B ELISAs are presented in Figure 2.

### 3.3. Distribution of Serological Findings by Abattoir, Sex, and Pig Breed

Pigs from many districts were processed at each, and abattoirs were not restricted to pigs from certain districts. The range in seropositivity against the OPS-A antigen in pigs across the six different slaughterhouses was between 3.74% (95% CI: 1.2–9.5%) at Budo and 9.22% (95% CI: 6.6–12.8%) at Lusanja using the 2 SD cutoff values. Seropositivity against the OPS-B antigen was between 3.85% (95% CI: 0.9–11.2%) at Buwate and 12.58% (95% CI: 9.3–16.8%) at Wambizi using 2 SD cutoff. For samples that were seropositive against both OPS-A and OPS-B antigens, positivity was between 1.28% (95% CI: 0–7.6%) at Buwate and 4.84% (95% CI: 2.9–7.9%) at Wambizi. At the 3 SD cutoffs, seropositivity against OPS-A antigen was between 0.00% at Budo and 2.90% (95% CI: 1.5–5.5%) at Wambizi. For the OPS-B antigen at 3 SD, the lowest seropositivity was at 0.00% at Buwate, and the highest was 4.84% (95% CI: 2.9–7.9%) at Wambizi. At Kyetume-Mukono, seropositivity against both antigens was 4.17% (95% CI: 0.4–14.8%) using the 3D cutoff, while Budo, Buwate, Katabi-Entebbe, and Lusanja abattoirs had no pigs positive to both.

Analysis of serological findings by sex showed that female pigs represented higher seropositivity to both ELISAs than male pigs. For the OPS-A ELISA at the 2 SD cutoff, the seropositivity among females was 10.07% (95% CI: 7.8–12.8%) and 3% (95% CI: 1.8–4.8%) at the 3 SD cutoff. Among females, seropositivity to the OPS-B ELISA was 10.07% (95% CI: 7.8–12.8%) at the 2 SD cutoff and 3.53% (95% CI: 2.3–5.4%) at the 3 SD cutoff. Among the males, 3.89% (95% CI: 2.4–6.1%) were positive for type A ELISA at the 2 SD cutoff and only 0.43% (95% CI: 0.01–1.7%) at the 3 SD cutoff. For the type B ELISA, 7.78% (95% CI: 5.6–10.6%) males were seropositive at the 2 SD cutoff and 1.73% (95% CI: 0.8–3.4%) at 3 SD. At 2 SD, the sex of pigs was significantly associated with exposure to *B. pseudomallei* Type A, *p*-value < 0.001, and to both type A and B, *p*-value = 0.006. However, it was not significantly associated with exposure to *B. pseudomallei* type B alone, with *p*-value = 0.201. At 3 SD, pig sex was significantly associated with exposure to *B. pseudomallei* type A, *p*-value = 0.002. However, there was no significant association between pig sex and exposure to type B alone, *p*-value = 0.077, or to both type A and B, *p*-value = 0.260.

Seropositivity varied by pig breed, and some pigs were seropositive to both type A and type B ELISAs. At the 2 SD cutoff, mixed breed pigs had 10.53% (95% CI: 7.4–14.7%) positivity to the OPS-A ELISA and 2.46% (95% CI: 1.1–5.1%) at 3 SD. For the OPS-B ELISA, mixed breed pigs had 14.39% (95% CI: 10.8–19%) seropositivity at the 2 SD cutoff and 4.91% (95% CI: 2.9–8.1%) at 3 SD. For the exotic pigs, seropositivity to the OPS-A ELISA at the 2 SD cutoff was 5.81% (95% CI: 4.1–8.1%) and 1.23% (95% CI: 0.5–2.6%) at the conservative 3 SD cutoff. For the OPS-B ELISA, 6.87% (95% CI: 5–9.3%) of exotic pigs were seropositive at the 2 SD cutoff and 2.11% (95% CI: 1.2–3.7%) at 3 SD. For local breed pigs, seropositivity to OPS-A ELISA at the 2 SD cutoff was 6.17% (95% CI: 3.3–11.1%) and 2.47% (95% CI: 0.7–6.4%) at the conservative 3 SD. Among local pigs, seropositivity to OPS-B ELISA was 8.02% (95% CI: 4.6–13.3%) at the 2 SD cutoff and 1.23% (95% CI: 0.05–4.7%) at 3 SD. At 2 SD, pig breed was significantly associated with *B. pseudomallei* type A, *p*-value = 0.036, and type B alone, *p*-value = 0.001. However, it was not significantly associated with exposure to both types A and B, with *p*-value = 0.256. At 3 SD, the pig breed was only significantly associated with *B. pseudomallei* type B, *p*-value = 0.027. However, it was not significantly associated with exposure to *B. pseudomallei* type A, *p*-value = 0.338, or to both type A and B, *p*-value = 0.913. A detailed breakdown of the serological findings by abattoir, sex of pig, and breed is presented in Table 1.

### 3.4. Seropositivity at the District Level

The effect of EBS smoothing is illustrated in box plots in Figure 3. The spatial distribution of raw and EBS seroprevalence is presented in Figure 4. Raw seroprevalence to either OPS-A or OPS-B at the 3 SD cutoff ranged from 0–33.3% (Figure 4A), while smoothed rates ranged from approximately 3% to 6.35% (Figure 4B). The Highest EBS rates were reported from the districts of Bukomansimbi, Kalungu, Mpigi (along the northern shore of Lake Victoria, Kampala (central to the abattoirs), Nakaseke (one of the central most districts sampled), and Iganga. The general patterns of seroprevalence were similar between raw and smoothed rates, though several of the highest EBS districts were underestimated in raw prevalence calculations, with Kiryandongo in the north of the sampled districts occurring in a higher prevalence category when smoothed.

### 3.5. Local Spatial Autocorrelation of EBS Seroprevalence

Local Moran’s I of EBS seroprevalence detected two high-high spatial clusters in Masaka and Kalungu (Figure 5A,B, respectively), both on the northwestern shore of Lake Victoria and a high-low spatial outlier in Nakaseke in the central areas of sampling (Figure 5C). 

### 3.6. Temporal and Seasonal Variation in Seropositivity

Higher absorbance in both the OPS-A and OPS-B assays was significantly associated with the rainy seasons (Figure 6A,B). High absorbance values in December could be due to the lag effects of rainfall from November or the rainy season continuing into the early part of December. Seropositivity was higher in the months of October, November, and December for both the OPS-A ELISA and OPS-B ELISA at the 2 SD and 3 SD cutoffs and particularly lower in the dry months of January, February, June, July, and August. For the OPS-A ELISA at the 2 SD cutoff, the seropositivity for October was 18.1% (95% CI: 11.8–26.6%) and 3.81% (95% CI: 1.2–9.7%) at the 3 SD cutoff. For the OPS-B ELISA at the 2 SD cutoff, the seropositivity for October was 19.05% (95% CI: 12.6–27.7%) and 5.71% (95% CI: 2.4–12.2%) at the 3 SD cutoff. For November and December, the seropositivity for the OPS-A ELISA at the 2 SD cutoff was 23.47% (95% CI: 16.1–32.8%) and 23.08% (95% CI: 16–32.1%), respectively. The seropositivity for November and December for the OPS-B ELISA at the 2 SD cutoff was 34.69% (95% CI: 26–44.6%) and 23.08% (95% CI: 16–32.1%), respectively. At the 3 SD cutoff, seropositivity for November and December for the OPS-A ELISA was 4.08% (95% CI: 1.3–10.4%) and 11.54% (95% CI: 6.6–19.2%), respectively. For the OPS-B ELISA at 3 SD, the November and December seropositivity was 12.24% (95% CI: 7–20.3%) and 7.69% (95% CI: 3.7–14.7%), respectively. A detailed temporal distribution of seropositivity is provided in Appendix A.

## 4. Discussion

To the best of our knowledge, this study is the first to document *B. pseudomallei* seropositivity in animals in East Africa. Only one preliminary study in humans was conducted in Uganda from 1971 to 1972 and reported a seroprevalence of 5.9% [27]. Being an environmental pathogen, exposure of pigs to this pathogen could suggest exposure to other livestock species and humans living in the locales where the sampled pigs were raised. Animals are known sentinels of disease and other public health hazards [37], and pigs have been used as sentinels for pathogens such as *Mycobacterium bovis* [38]. 

The findings suggest that pig slaughterhouse workers and meat inspectors could be exposed to *B. pseudomallei* while handling infected pig carcasses. Transmission of this pathogen to humans through the handling and processing of infected animals is believed to occur [39]. Persons consuming contaminated or undercooked pork could as well be exposed because ingestion of contaminated food is an important risk factor for exposure [40]. Awareness of Ugandan pig slaughterhouse workers about infection prevention and control and hygiene practices to minimize meat contamination while working at the slaughterhouses is necessary. Our study also suggests possible differences in seropositivity among female and male pigs as well as among the different breeds of pigs in Uganda. These differences (if any) might be associated with differences in pig husbandry practices in the country.

In the present study, pigs originating from at least 43 districts in Uganda had serological evidence of exposure to *B. pseudomallei*, suggesting that this pathogen could be widely distributed in Uganda (38 samples did not list district). Assays were run targeting exposure to both type A and type B *B. pseudomallei*. The lipopolysaccharide type A or type B sugars that coat the surface of the organism can cause different serological reactions [41,42]. It is currently unknown whether type A or B LPS *B. pseudomallei* predominate in soils across Africa. A phylogenomic study of a limited subset of *B. pseudomallei* from Madagascar found two-thirds of strains were type B LPS [43]. In the present study, the performance of the tests indicates exposure to both type A and type B LPS *B. pseudomallei* with some animals seropositive to both OPS-A and OPS-B assays using a conservative cutoff. This provides indirect evidence that type A and type B LPS *B. pseudomallei* are present in Uganda. Isolation and typing of viable *B. pseudomallei* are required for a definitive answer. 

Smoothed district-level rates ranged from approximately 3–6%, which is in line with recent estimates for swine in Vietnam, where human melioidosis epidemiology is better studied [28]. In Uganda, we identified spatial variation across districts, with several having relatively high rates; two districts along the shore of Lake Victoria were part of a high-high cluster, and one was a spatial high-low outlier. High-high clusters include districts with the highest predicted rates and neighbors that also have higher rates compared to the mean. High-low outliers have high predicted prevalence with neighbors that have low predicted prevalence. This cluster and outlier can be interpreted as hotspots of pig exposure to *B. pseudomallei* within Uganda. Each of these three districts has flooding rivers, delta areas, or lake shores, all wet areas which are associated with *B. pseudomallei* isolation [44,45,46]. Widespread detection is not unexpected, as *B. pseudomallei* is an environmental pathogen that is predicted to be ubiquitous in tropical regions [21]. Identification of regional hotspots can provide a starting point for understanding the environmental burden of melioidosis in Uganda. In areas of high seroprevalence, we can target efforts to identify the human and animal health impacts of *B. pseudomallei* exposure. We also need to interpret the data carefully. Mpigi and Bukomansimbi were additional areas with high *B. pseudomallei* exposure in pigs, given the relative seropositivity rates observed there. It is important to note the central region of Uganda is known to have a high pig density [47]. Likewise, the Iganga district in eastern Uganda and Kiryandongo in the west are also areas of interest for future studies. However, more robust sampling and surveillance are needed for to better understand the epidemiology of *B. pseudomallei* in Uganda.

Seasonal variation in the occurrence of melioidosis, a disease caused by *B. pseudomallei,* has been documented by several studies, and in tropical Australia, its occurrence has predominantly been associated with the wet seasons [16,48,49]. Our findings suggest a higher level of exposure of pigs to this pathogen in the rainy months. It is assumed that high-risk persons such as farmers, veterinary staff, and slaughterhouse workers in Uganda would have a higher level of exposure during the rainy seasons. However, this needs to be investigated.

Strict COVID-19 lockdown measures in Uganda hampered sampling, particularly during the months of June and July 2021. These COVID-19 lockdown measures limited the movement of people, which could have affected pig trader movements and the number of animals sent to the slaughterhouses. Hence, selection bias is likely an issue in this study. The statistical procedure for identifying the seropositive samples is not robust. This is a known limitation of many serological assays; however, it allows estimates of disease prevalence, a necessary first step in incidence validation [30]. Despite the limitations, this proof-of-concept study provides evidence of the likely endemicity of *B. pseudomallei* in East Africa. Our next step is to deeply characterize the distribution of this pathogen in the East African region at the One Health interface (in humans, animals, and the environment).

## 5. Conclusions

The findings of this pilot study show that *Burkholderia pseudomallei* is present in Uganda and appears prevalent in pigs from districts representing all regions of the country. Both *B. pseudomallei* type A and type B were detected in the study. Pig slaughterhouses in the Kampala metropolitan area of Uganda slaughter pigs exposed to *B. pseudomallei*. This pathogen appears to be endemic and neglected in Uganda, suggesting further study and expanded surveillance are warranted. Public health awareness campaigns about the disease may also be needed. 

## Figures and Tables

**Figure 1 pathogens-11-01363-f001:**
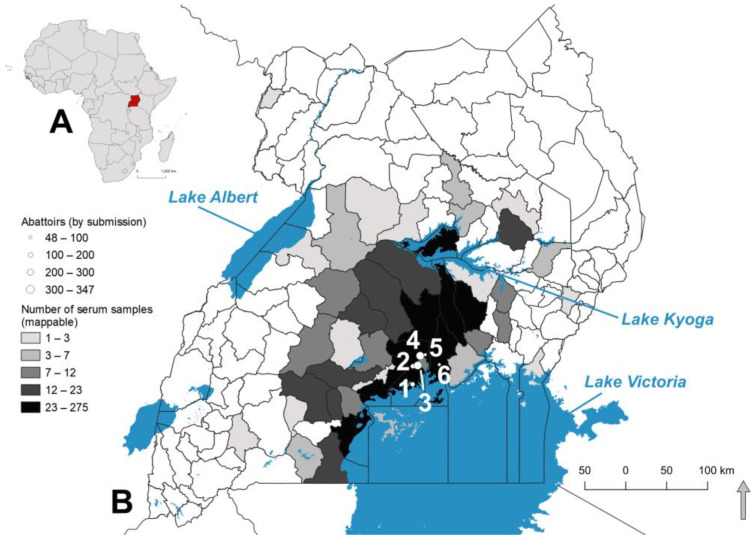
(**A**) Location of Uganda in Africa and (**B**) distribution of samples across districts in Uganda. Districts are choropleth mapped by the number of samples per submission. Samples were collected across six abattoirs (white points): Katabi (1), Budo (2), Wambizi (3), Lusanja (4), Buwate (5), and Kyetume (6). Abattoirs are graduated by size per number of pigs sampled.

**Figure 2 pathogens-11-01363-f002:**
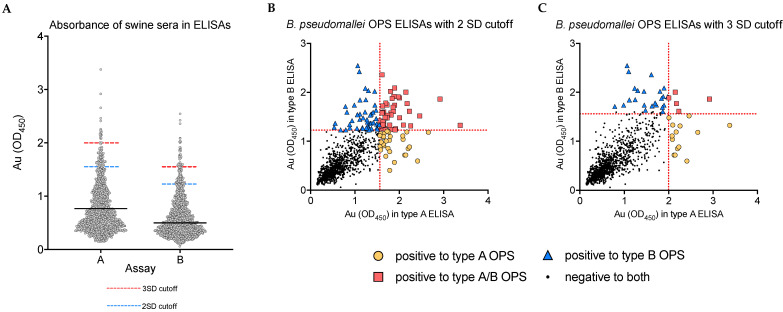
Raw absorbance in the OPS-A and OPS-B ELISAs (**A**) Absorbance of all pig samples to the OPS type A and OPS type B ELISAs. Black lines indicate the average absorbance values in each assay, with cutoffs at three standard deviations (SD) and two standard deviations above the mean indicated by dashed red and blue lines, respectively. (**B**) Absorbance of samples in the OPS-A ELISA (X-axis) plotted against the absorbance in the OPS-B ELISA (Y-axis) with 2 SD cutoffs represented by dashed red lines at Au = 1.561 and Au = 1.226, respectively. (**C**) Absorbance of samples in the OPS-A ELISA (X-axis) plotted against the absorbance in the OPS-B ELISA (Y-axis) with 3 SD cutoffs represented by dashed red lines at Au = 1.998 and Au = 1.561, respectively. In (**B**,**C**), samples passing the cutoff in the OPS-A ELISA are the yellow circles, samples passing the cutoff in the OPS-B ELISA are blue triangles, and those samples passing the cutoff in both ELISAs are red squares. Samples falling below cutoffs in both assays are represented with black dots.

**Figure 3 pathogens-11-01363-f003:**
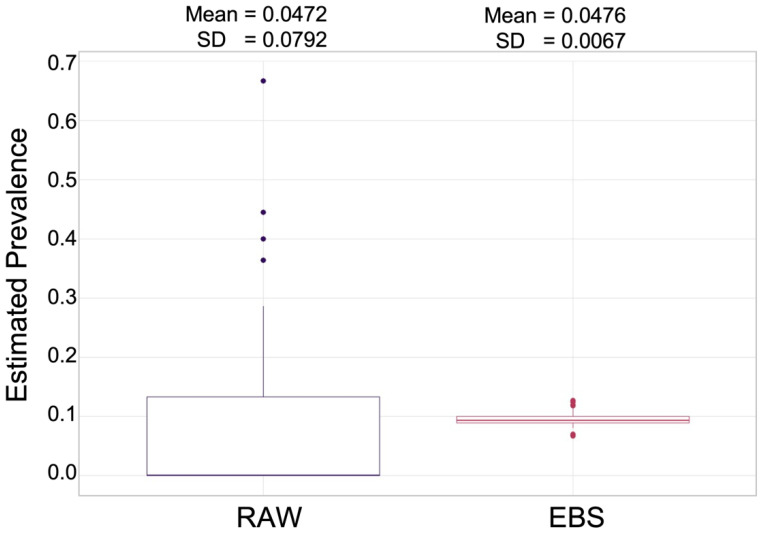
Box plots of estimated seroprevalence for districts using raw (left; purple box) and empirically Bayes smoothing (EBS; right; red box) performed in GeoDa 1.20. The mean and SD of the data sets are indicated above the box plots.

**Figure 4 pathogens-11-01363-f004:**
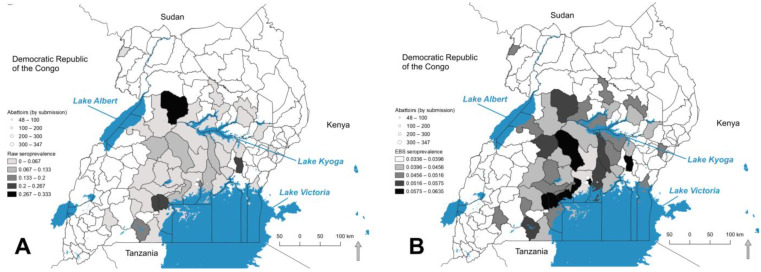
(**A**) Raw seroprevalence rates of *B. pseudomallei* exposure in pigs by district. (**B**) Empirically Bayes smoothed (EBS) seroprevalence rates per district. Positive was defined as the animal having OD values at the three standard deviation cutoffs for either OPS-A or OPS-B. Samples below the 3 SD cutoffs for either test were considered negative. White points represent the abattoirs where samples were submitted; mapped data excluded 38 cases lacking district of origin.

**Figure 5 pathogens-11-01363-f005:**
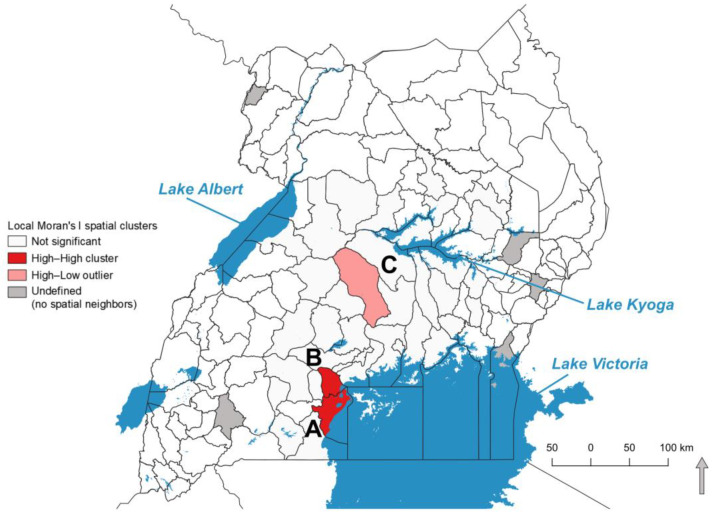
Local Moran’s I-defined spatial clusters of *B. pseudomallei* seroprevalence for pigs tested in Uganda. High–High clusters were identified in (**A**) Masaka and (**B**) Kalungu, both bordering Lake Victoria. A single spatial outlier (High–Low) was identified in (**C**) Nakaseke, a district bordered by rivers.

**Figure 6 pathogens-11-01363-f006:**
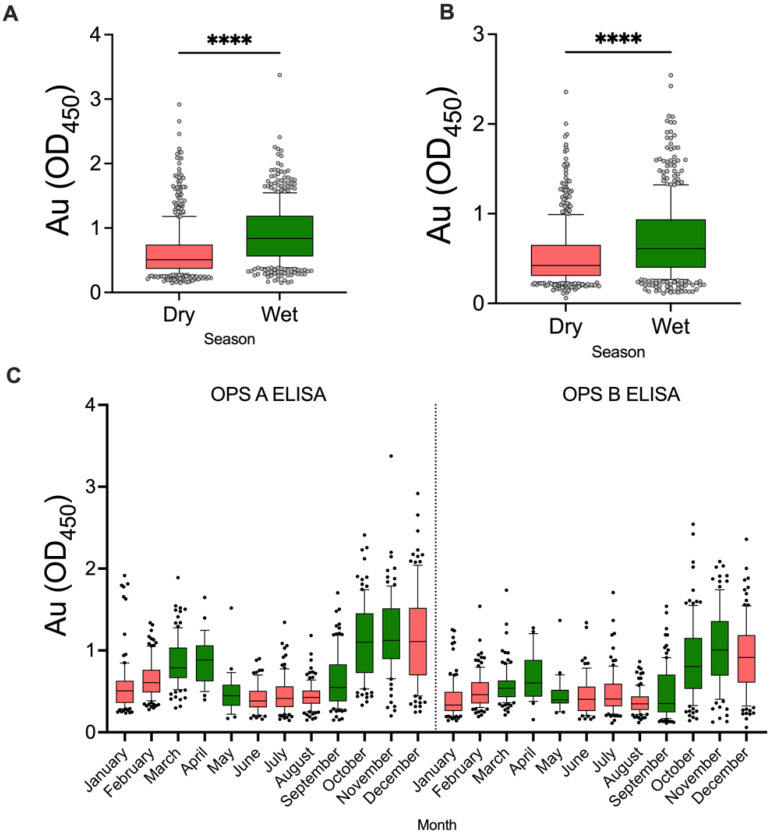
Absorbance data organized by rainfall season and month. (**A**) Sample absorbance data organized by dry and wet seasons to the OPS-A assay. (**B**) Sample absorbance data organized by dry and wet seasons to the OPS-B assay. (**C**) Sample data was organized by month of collection. Months considered ‘wet’ are colored green, while ‘dry’ months are colored red. The mean and interquartile ranges of absorbance values are indicated by the boxes, with the 10th and 90th percentiles indicated by whiskers. Outliers are indicated by dots. ****, *p* < 0.0001 by the Mann-Whitney U test.

**Table 1 pathogens-11-01363-t001:** Abattoir, sex, and pig breed specific unadjusted prevalence estimates with 95% confidence intervals (CI) at each of the two and three standard deviation cutoffs, respectively (2SD, 3SD).

	2 Standard Deviation Cutoff						3 Standard Deviation Cutoff						
Abattoir (District)	# Positive A	% (95% CI) Positive A	# Positive B	% (95% CI) Positive B	# Positive Both A & B	% (95% CI) Positive Both A & B	# Positive A	% (95% CI) Positive A	# Positive B	% (95% CI) Positive B	# Positive Both A & B	% (95% CI) Positive Both A & B	Total
Budo (Wakiso)	4	3.74% (1.2–9.5%)	8	7.48% (3.6–14.3%)	3	2.80% (1.6–8.3%)	0	0.00%	2	1.87% (0.1–7%)	0	0.00%	107
Buwate (Wakiso)	4	5.13% (1.6–12.8%)	3	3.85% (0.9–11.2%)	1	1.28% (0–7.6%)	1	1.28% (0–7.6%)	0	0.00%	0	0.00%	78
Katabi (Wakiso)	8	5.52% (2.7–10.7%)	13	8.97% (5.2–14.8%)	7	4.83% (2.2–9.8%)	1	0.69% (0–4.2%)	4	2.76% (0.8–7.1%)	0	0.00%	145
Lusanja (Wakiso)	32	9.22% (6.6–12.8%)	27	7.78% (5.4–11.1%)	16	4.61% (2.8–7.4%)	7	2.02% (0.9–4.2%)	5	1.44% (0.5–3.4%)	0	0.00%	347
Kyetume (Mukono)	2	4.17% (0.4–14.8%)	3	6.25% (1.5–17.5%)	2	4.17% (0.4–14.8%)	1	2.08% (0–11.9%)	2	4.17% (0.4–14.8%)	1	2.08% (0–11.9%)	48
Wambizi (Kampala)	25	8.06% (5.5–11.7%)	39	12.58% (9.3–16.8%)	15	4.84% (2.9–7.9%)	9	2.90% (1.5–5.5%)	15	4.84% (2.9–7.9%)	4	1.29% (0.4–3.4%)	310
Total	75	7.25% (5.8–9%)	93	8.99% (7.4–10.9%)	44	4.25% (3.2–5.7%)	19	1.84% (1.2–2.9%)	28	2.71% (1.9–3.9%)	5	0.48% (0.2–1.2%)	1035
**Sex of Pigs**													
Female	57	10.07% (7.8–12.8%)	57	10.07% (7.8–12.8%)	33	5.83% (4.2–8.1%)	17	3.00% (1.8–4.8%)	20	3.53% (2.3–5.4%)	4	0.71% (0.2–1.9%)	566
Male	18	3.89% (2.4–6.1%)	36	7.78% (5.6–10.6%)	11	2.38% (1.3–4.3%)	2	0.43% (0–1.7%)	8	1.73% (0.8–3.4%)	1	0.22% (0–1.3%)	463
Unknown	0	0.00%	0	0.00%	0	0.00%	0	0.00%	0	0.00%	0	0.00%	6
Total	75	7.25% (5.8–9%)	93	8.99% (7.4–10.9%)	44	4.25% (3.2–5.7%)	19	1.84% (1.2–2.9%)	28	2.71% (1.9–3.9%)	5	0.48% (0.2–1.2%)	1035
**Breed of Pigs**													
Mixed	30	10.53% (7.4–14.7%)	41	14.39% (10.8–19%)	17	5.96% (3.7–9.4%)	7	2.46% (1.1–5.1%)	14	4.91% (2.9–8.1%)	1	0.35% (0–2.2%)	285
Exotic	33	5.81% (4.1–8.1%)	39	6.87% (5–9.3%)	22	3.87% (2.5–5.8%)	7	1.23% (0.5–2.6%)	12	2.11% (1.2–3.7%)	3	0.53% (0.1–1.6%)	568
Local	10	6.17% (3.3–11.1%)	13	8.02% (4.6–13.4%)	5	3.09% (1.1–7.2%)	4	2.47% (0.7–6.4%)	2	1.23% (0.1–4.7%)	1	0.62% (0–3.8%)	162
Unknown	2	10.00% (1.6–31.3%)	0	0.00%	0	0.00%	1	5.00% (0–25.4%)	0	0.00%	0	0.00%	20
Total	75	7.25% (5.8–9%)	93	8.99% (7.4–10.9%)	44	4.25% (3.2–5.7%)	19	1.84% (1.2–2.9%)	28	2.71% (1.9–3.9%)	5	0.48% (0.2–1.2%)	1035

# = number.

## Data Availability

The datasets used and/or analyzed during the current study are available from the corresponding authors upon reasonable request.

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
