# Peer review of "An Investigation of Burkholderia pseudomallei Seroprevalence in Market Pigs Slaughtered at Selected Pig Abattoirs in Uganda"

_pathogens, 2022, doi:10.3390/pathogens11111363_

Round 1
Reviewer 1 Report
In this work the authors present a large collection of sera samples from animals used for human consumption to evaluate the prevalence of the human pathogen Burkholderia pseudomallei. The manuscript is well organized and the data presented a good quality. However, I considere that discussion section could be improved adding for example the risk of ingestion of contaminated food, as recently Bpm has been described as gastrointestinal pathogen in animal models (ref: 10.1128/IAI.00654-20; 10.1093/trstmh/trx031).
Minor:
- Lines 61-63: also because it can be used in aerosol and has low infectious rate
- Line 74-76: Sporadic human melioidosis cases have been reported in Uganda [27], proving its existence, while the extent of the disease in both animals and humans remains unknown [22].
Line 109: (PBST)
Line 111: 1 hour
Line 113: incubated (lower case)
Line 114: inverted (lower case)
Line 133: For each antigen tested by ELISA,
Line 178: Seroprevalence
Line 264: B. pseudomallei (italics)
Page 9 line 6: (along the northern shore of Lake Victoria)
Figure 6: I think the significance is missing in graphs, while explained in the footnote
Author Response
In this work the authors present a large collection of sera samples from animals used for human consumption to evaluate the prevalence of the human pathogen Burkholderia pseudomallei. The manuscript is well organized and the data presented a good quality. However, I considered that discussion section could be improved adding for example the risk of ingestion of contaminated food, as recently Bpm has been described as gastrointestinal pathogen in animal models (ref: 10.1128/IAI.00654-20; 10.1093/trstmh/trx031).
Response: We thank the reviewer for the time spent reviewing our manuscript and for sharing the paper describing Bpm as a gastrointestinal pathogen in animal models. The review feedback has helped us improve our manuscript. We have added ingestion of contaminated food as a risk factor for exposure to Bpm and cite the paper that was shared by the reviewer. Please see page 11, lines 67 to 77.
Minor:
- Lines 61-63: also because it can be used in aerosol and has low infectious rate
Response: We thank you so this suggestion. Indeed, the pathogen can be aerosolized and has a low infectious rate. We have added this information and added an appropriate citation. Please see page 2, lines 60-64 in the revised manuscript.
- Line 74-76: Sporadic human melioidosis cases have been reported in Uganda [27], proving its existence, while the extent of the disease in both animals and humans remains unknown [22].
Response: We have revised the statement by replacing “however” with “while”. Please see lines 76 in page 2 of the revised manuscript.
Line 109: (PBST)
Response: We believe PBS is correct here because the sentence states Tween in PBS. I have put PBST in parenthesis to indicate the abbreviation. Please see page 3, lines 110-111.
Line 111: 1 hour
Response: We have edited the text by replacing 1-hour with 1 hour. Thank you. Please see line 112 in page 3 of the revised manuscript.
Line 113: incubated (lower case)
Response: Thank you for the feedback. We have corrected this typo. Please see line 114 in page 3 of the revised manuscript.
Line 114: inverted (lower case)
Response: We thank you for the feedback. We have corrected this typo. Please see page 3, line 115 in the revised manuscript.
Line 133: For each antigen tested by ELISA,
Response: We thank you for this suggestion. We have revised the statement to reflect the suggestion by the reviewer. Please see line 134 in page 3 of the revised manuscript.
Line 178: Seroprevalence
Response: We thank you pointing out this typo. We have corrected it. Please see page 4, line 179 in the revised manuscript.
Line 264: B. pseudomallei (italics)
Response: We have italicized “B. pseudomallei” as suggested. Thank you for this. Please see line 265 in page 6 of the revised manuscript.
Page 9 line 6: (along the northern shore of Lake Victoria)
Response: We thank you for the feedback. We have corrected this typo. Please see page 9 line 6 in the revised manuscript.
Figure 6: I think the significance is missing in graphs, while explained in the footnote
Response: We thank the reviewer for this observation. We have added asterisks in the graphs “****” that were missing in Figure 6A and B and replaced Figure 6 with a png to solve the problem. Please see Figure 6.
Other revisions
Line 138, page 3: We changed “Organization” to “Organisation” because that is how it is spelled in the WOAH website.
Line 169, page 4: Deleted “rates”, because prevalence is not a rate, but a proportion.
Reviewer 2 Report
The idea of study the prevalence of B. pseudomallei as a pathogen responsible for melioidosis on slaughetered pig, as well as the recognization of hot spots on Uganda is noteworthy as it can be used on the development of better health strategies on health programs. The paper seems to be interesting and wanted. The research is well designed and performed. The article is clearly written and the presentation style is appropriate for a scientific journal.
Broad comments: Figure 2. Title should be improved, first indicate the tittle and later the subfigure letters.-Figure 3. could be supplementary
in general the frequency of some serum type B. pseudomallei (A or B) is significative, the conclusions ubicated as a hot spot the Kampala metropolitan area of Uganda? please clarify and be more descriptive about the significant percentage for consider a hot spot zone.
.in conclusions could emphasize the perspective for the obtained results
Author Response
The idea of study the prevalence of B. pseudomallei as a pathogen responsible for melioidosis on slaughetered pig, as well as the recognization of hot spots on Uganda is noteworthy as it can be used on the development of better health strategies on health programs. The paper seems to be interesting and wanted. The research is well designed and performed. The article is clearly written and the presentation style is appropriate for a scientific journal.
Response: : We thank the reviewer for the time spent reviewing our manuscript and for providing feedback that has greatly improved our manuscript.
Broad comments: Figure 2. Title should be improved, first indicate the tittle and later the subfigure letters.
Response: We thank you for this suggestion. We have improved the title of Figure 2 by first indicating the title and then providing the subfigure letters. Please see page 6, line 229.
-Figure 3. could be supplementary
Response: We thank the reviewer for this suggestion. However, we aim to leave Figure 3 in the main text because it is required as it differs from the mapped results in what it describes. As we mention in lines 169-170, Figure 3 is required because it gives a description of the effect of smoothing on prevalence.
in general the frequency of some serum type B. pseudomallei (A or B) is significative, the conclusions ubicated as a hot spot the Kampala metropolitan area of Uganda? please clarify and be more descriptive about the significant percentage for consider a hot spot zone.
Response: We thank the reviewer for this comment. To clarify, the pig serum samples were collected from pigs slaughtered at six pig abattoirs located in the Kampala metropolitan area of central Uganda. However, we collected data of the district of origin of the pigs which we used for our spatial cluster analysis to identify B. pseudomallei hotspots. So, the samples were from pigs coming into Kampala metropolitan area from different districts. We added the words “in the Kampala metropolitan area” to lines 83-86 in page 2 of the revised manuscript for purposes of clarity.
To identify the spatial clusters (hotspots), we used Local Moran’s I statistic in GeoDa—a spatial data analysis software and we provide reference 33. We describe this in our manuscript in section “2.5. Local Spatial Autocorrelation of Seroprevalence” (lines 179-202). We are careful not to use raw prevalence data to suggest hotspots because raw rates run the risk of identifying clusters where they may not be real due to high sample variance.
.in conclusions could emphasize the perspective for the obtained results
Response: We thank the reviewer for this comment. We have reworded the opening statement of the conclusions to emphasize the perspective of our results. Please see page 13, lines 130-132 in the revised manuscript.
Other revisions
Line 138, page 3: We changed “Organization” to “Organisation” because that is how it is spelled in the WOAH website.
Line 169, page 4: Deleted “rates”, because prevalence is not a rate, but a proportion.